# Merging Observed and Self-Reported Behaviour in Agent-Based Simulation: A Case Study on Photovoltaic Adoption

**Andrea Borghesi \*** and **Michela Milano**

DISI, University of Bologna, 40136 Bologna, Italy; michela.milano@unibo.it
\* Correspondence: andrea.borghesi3@unibo.it

**Abstract:** Designing and evaluating energy policies is a difficult challenge because the energy sector is a complex system that cannot be adequately understood without using models merging economic, social and individual perspectives. Appropriate models allow policy makers to assess the impact of policy measures, satisfy strategic objectives and develop sustainable policies. Often the implementation of a policy cannot be directly enforced by governments, but falls back to many stakeholders, such as private citizens and enterprises. We propose to integrate two basic cornerstones to devise realistic models: the self-reported behaviour, derived from surveys, and the observed behaviour, from historical data. The self-reported behaviour enables the identification of drivers and barriers pushing or limiting people in their decision making process, while the observed behaviour is used to tune these drivers/barriers in a model. We test our methodology on a case-study: the adoption of photovoltaic panels among private citizens in the Emilia–Romagna region, Italy. We propose an agent-based model devised using self-reported data and then empirically tuned using historical data. The results reveal that our model can predict with great accuracy the photovoltaic (PV) adoption rate and thus support the energy policy-making process.

**Keywords:** simulation model; multi-agent systems; photovoltaic energy; parameter fine-tuning; self-reported behaviour; predictive model

---

## 1. Introduction

The European Union is deeply committed to curtailing its greenhouse gas emissions by at least 20% by 2020, w.r.t. 1990 levels, as stated in the sustainable growth strategy outlined in [1]. The path to achieve such a goal passes through an increase up to 20% of the share of renewable energy sources in final energy consumption and a 20% rise in energy efficiency. All EU members and regions should put an effort in this direction to contribute to these common objectives. For instance, Italy was supposed to reach a 17% share of final energy coming from renewable sources in 2014, a target that have been reached and slightly surpassed [2].

The complex task of enforcing these guidelines is shouldered by national and regional policy makers. Energy policies have a strong impact on sustainable development and they influence economy, society and environment. Policy makers have to devise plans targeting strategic objectives, e.g., cutting greenhouse emissions, with the goal of satisfying different constraints (i.e., limiting pollutant emissions, not exceeding a financial budget, etc.) and respect the EU guidelines. After having been devised, the plans need to be enforced with implementation instruments (from incentives to investment grants, passing through tax exemptions) [3–5]. One aspect that tends to be severely underestimated while planning energy policies is the strong influence of human behaviour together with social dynamics; it is often studied with the assumption that consumers are rational and guided only by financial and

economic drivers [6–8] which severely affect the accuracy and realism of the study. In fact, the decision process of involved agents (i.e., private citizens) is deeply influenced by non-economical motivations, such as social influence, peer pressure, bandwagon effects, lack or wealth of knowledge, risk aversion, etc. [9–12]. Properly understanding the decision making process is critical to better influence the interested parties' behaviour and steering them toward good practices and policy objectives. In this context there is urgent need of appropriate and accurate models for enabling policy makers to design, evaluate and implement energy policies to satisfy strategic objectives and develop sustainable strategies that have a strong impact on economy, society and environment.

We propose to merge in the model definition two types of knowledge: (1) self-reported behaviour derived from large scale surveys and interviews, and (2) observed behaviour based on real data measuring the actual effect of the target energy policy. The models are used to bridge the gap between these two behaviours and enable a better understanding of private citizens decision-making processes. We claim that social and economic drivers and barriers can be extracted from quantitative analysis of survey data, whilst a deeper understanding of how these drivers operate and interact can be derived from interview findings. On the basis of these drivers and barriers, we build a parametric model, whose parameters can be empirically tuned so that the model reproduces the observed behaviour. We expect the parameter tuning to generate different outputs (i.e., parameter values for different drivers and barriers) for different private entities classes (private citizens, enterprises, etc.) and for different countries and geographical situations.

The final outcome of merging self-reported and observed behaviour is the creation of predictive models with the ability to forecast the stakeholder behaviour in the presence of specific energy policies, financial and economic situations. These predictive models can be inserted into simulations and used by policy makers in a what-if fashion, namely by proposing alternative scenarios and observing the emerging behaviour of consumers related to energy efficiency and overall cost.

In this work we focus on policies for promoting energy production from renewable energy sources and, in particular on photovoltaic (referred to as PV) power generation. We use, as a case study, self-reported and observed behaviour in the Italian region of Emilia–Romagna, where the majority of the total installed photovoltaic power is generated by small/medium panels installed by private citizens and enterprises. For this reason the regional policy makers cannot directly decide the total power installed, but they have to push the PV power generation through indirect means, usually in the form of incentives to the PV energy. The decision to install a PV panel is not driven exclusively by economical/technical considerations (although these aspects have clearly a significant impact), but it involves also different factors determined by the human behaviour and social interactions [13,14]. As observed behaviour, we employ the data regarding the historical yearly installation rate of new photovoltaic panels and the total amount of installed photovoltaic power reported by the national and regional governments. On these data we craft an agent-based model for simulating the adoption of photovoltaic panels. We consider individual households as the actors populating the simulation environment and deciding whether to install a PV panel or not. The behavioural rules of the agents are devised using self-reported data collected thanks to surveys and questionnaires conducted among private citizens. From these data we derive the drivers and the barriers that influence the adoption of a PV panel. The importance of each factor is decided during the following phase, when we use the observed data in order to fine-tune the parameters of the model. The model takes into account both geographical, economical and social aspects.

The validation and final evaluation of the proposed model has been performed over a period of 11 years by comparing the historical PV power installation trend in a certain period to the one generated by the agent-based simulators. The historical data collected over this time span is divided in two sub sets in order to achieve a two-fold purpose: (I) tune the agent-based model's parameters (combination of self-reported and observed behaviour) and (II) test the accuracy of the approach by assessing its predictive capacity. For this purpose, the first seven years were used for parameter tuning and for the remaining years we compare the historical data with the simulated behavior—a small

discrepancy would mean a good accuracy of the model, otherwise, a large gap would indicate a model not really usable. The experimental results highlight that with our model it was possible to predict future trend of installed PV power; this information and predictive capability can greatly help policy makers in their task.

The structure of the paper is the following. In Section 2 related works are discussed. Section 3 provides a general overview of the proposed approach. Section 4 presents the surveys used to identify drivers and barriers governing people's decisions and the method to derive the model for the agents' behaviour. Then, Section 5 describes the proposed agent-based model. The method used for tuning the model's parameters is described in Section 6. Section 7 reports the evaluation of the proposed approach, validating the fine-tuned agent-based model and assessing its accuracy. Finally, Section 8 concludes the paper, summarizing the obtained results and suggesting future research directions.

## 2. Related Work

The adoption of renewable energy sources, such as photovoltaic panels, can be framed as an innovation diffusion problem, an issue that has been the subject of many research works. Several findings suggest that the diffusion of an innovation is a social process. A common methodology to deal with this problem is agent-based modeling and simulation, where the agents are connected to form a interconnected network; agent-based models are also referred in the literature (and in the rest of this paper) as ABMs. Agent-based modeling is a computational approach that provides a tool for researcher with the purpose of creating, analysing and experimenting with models composed of agents that interact within an environment. Agent-based models are a simplified representation of the reality that can be used to explore certain aspects that would be harder to study without the aid of computational experiments [15]. Agents are usually distinct parts of a program that are used to represent social actors/individuals, organizations such as firms and enterprises, or bodies such as nation-states. They are programmed to react to the computational environment where they reside; this "simulated" environment is a representation of the real environment where the social actors operate [16].

In particular, ABMs have been used to study how innovative technologies spread in the real world [17–20]. It has been noted that the adoption rate of innovation does not depend exclusively on economic factors (i.e., costs or available budget), but many other aspects can have a profound influence. For instance, Abrahamson et al. [21] describe a threshold ABM where the adoption rate of a new technology is influenced by the "bandwagon effect", with new adopters facilitating the spread of knowledge that in turn increases the adoption of the innovative technology by new agents. Similarly, Chatterjee et al. [22] consider that potential adopters can have precise information about the cost of a innovative technology but can only estimate its benefits and real value—hence the perceived worthiness is an important factor. The main idea is that the information about an innovative technology spreads among an increasing network of agent through communication with previous adopters—in this way the uncertainty about the innovation potential decreases. The PV technology diffusion can be cast as a problem of innovation adoption, hence these insights will be partially incorporated in the model proposed in this paper.

Extensive research has been devoted to investigating the PV technology via ABMs [23–26], with a special focus on rooftop PV panels—systems that are typically installed by private citizens and small enterprises. The rest of this section reviews some of the approaches proposed in the last few years.

Zhao et al. [27] describe an ABM for studying the diffusion of PV systems where agents are homeowners which decide whether to install a PV panel or not. They consider four main factors that affect the agents' decision: payback period, household income, neighbourhood and advertisement. The final decision of each agent is based on a linear combination of these four factors, called "desire level". If the desire level is above a certain threshold, the household will opt for installing a PV panel. Selecting the correct value for the threshold is not an easy task: it strongly impacts the decision algorithm of the agents but the authors do not offer a general method to compute it. Instead, the domain

experts' knowledge is used to select a set of realistic values, which have to be tested and validated on test-case scenarios (without comparison with historical trends). With our approach we aim at finding the correct values for the ABM parameters through the fine-tuning process guided with observed data.

Extending the work of Zhao et al., Palmer et al. propose a different ABM [28]: again, the agent decision criterion is based on four different factors, but in this case these factors are weighted according to the agents' social class. Each agent (corresponding to a household) is associated to a specific social class; a small-world network connects all agents and households belonging to the same classes tend to be linked together. The model parameters are calibrated using the PV installation trend in Italy during the 2006–2011 period but all the data set is using for the training, therefore no validation is performed using new data. This poses a risk of overfitting the model parameters to the particular historical period taken into consideration. The risk is also increased due to the selected period: in 2006–2011 the PV installation rate was mainly governed by a set of incentives offered by the Italian government that changed considerably in 2012, as was described in detail by Borghesi et al. in [29].

The approaches listed so far discounted the geographical location of buildings. Robinson et al. [30] introduce this new element and integrate data coming from a geographic information system (GIS) with an ABM, with the goal of analysing the diffusion rate of PV panels. The addition of the actual topology of the target area permits to include the effects of solar exposure and population density on the diffusion of PV systems, thus improving the accuracy of the model. The parameters calibration is done using real data of the historical PV adoption in the city of Austin, Texas. While the results are interesting, no validation has been performed yet, i.e., all the data has been used to fine-tune the model, whose accuracy w.r.t. new observed behaviour has not been computed. Davidson et al. [31] take into account geo-spatial information as well as population demographics in order to forecast the photovoltaic adoption trends. Their goal consists in understanding the best predictors for the installation of PV panels. The analysis highlights that a relatively small subset of geo-spatial data can be used to obtain estimates (in terms of PV adoption trend) as accurate as those obtained with much larger and more comprehensive geo-spatial data sets. This work does not develop an agent-based model and it is mostly focused on understanding which are the geo-spatial factors with the major impact on the PV adoption and thus is not directly comparable with the one proposed in this paper.

Zhang et al. [32] outline an ABM to study the adoption (both at individual and community level) of rooftop PV panels, considering the San Diego county as a case study. The key point of the proposed approach is to learn a model for the behavior of individual agents using combined data of individual adoption characteristics and property assessment; the learned model is then integrated in the agents involved in the simulation. They also employ their system to evaluate different policy strategies targeted at fostering PV adoption. The proposed model is calibrated using observed PV adoption rates in the city San Diego, California. The authors also propose a preliminary validation of their model, comparing its prediction accuracy to the accuracy of baseline model (a model taking into account fewer factors than the presented one). The validation lacks a full comparison of the model predicted behavior and the observed one.

Macal et al. [33] describe an agent-based model (called BE-Solar), that incorporates a social and behavioral decision framework for understanding the technology adoption process. The main limitation of this approach is the lack of model calibration and validation. Rai et al. [34] present a empirically-driven agent-based model of technology adoption applied to residential solar photovoltaic. The variables describing the agents' behaviour are fine-tuned using historical data. In their work, they propose a theoretically-based framework and consider multiple validation criteria.

Agent-based models have also been employed to simulate policy scenarios and provide recommendations. For example, Lee et al. [35] proposed an ABM to model the decision-making process of homeowners while buying and installing energy efficient technologies in their homes. Homeowners' decisions are based on a simple additive weighting algorithm that estimates the utility values of different options, ultimately selecting the one with the maximum utility value. The utility

values of different options are calculated based on a combination of empirical factors (derived from housing stock data), social factors, and policy regulations. Installations lead to altered energy demand and $CO_2$ emissions. The model was partially calibrated using observed data; due to the limited availability of historical data only a couple of technologies were subjected to calibration. Although this is a clear limitation, validation was not the main scope of the paper that instead was mostly focused on providing a tool for comparing different scenarios.

Johnson et al. [36] model households photovoltaic solar panels adoption following an approach where household agents initially make decisions based on their subjective beliefs, awareness, and attitude towards the technology. These factors determine the chance the homeowners meet with a photovoltaic installation company, at which point they become rational profit-maximising consumers, weighing up the costs and benefits (subsidies etc.) of installing solar panels. This model enabled the researchers to make recommendations to regional government on the potential impact of incentive policies, and how different policies compared in terms of costs, energy capacity installed, and participation rates. No validation technique for this model was proposed, as this work is more focused on studying the theoretical impact of different policies, rather than providing a predictive tool.

Adepetu et al. [37] employ an ABM to study the impact of realistic incentive mechanisms on the adoption and diffusion of PV-batteries. They observe that while many different types of incentives have been proposed in the last few years, those incentives did not have the same effect in different parts of the world, due to the underlying different conditions and contexts (referred to as jurisdictions). Then, in their work they propose jurisdiction-specific ABMs in order to find the best incentive policies. As a case study they consider two distinct jurisdictions, Ontario and Germany. The agents' decision process is partially based on questionnaires (self-reported behaviour) and the tuning of some of the model's parameters is performed using historical data. The proposed approach is interesting but it is explicitly focused on PV-batteries and do not consider PV panels on rooftops, hence it is not directly comparable to ours; moreover, the authors validate the models using historical data (looking for parameters values that lead to the best fit with observed data) but they do not evaluate the predictive capacity of the proposed ABMs—the main focus is the comparison of incentive policies among different jurisdictions. Furthermore, the proposed paper do not provide statistical, quantitative measures of the accuracy of the fitting method, hence the comparison is not possible.

Sinitskaya et al. [38] describe an agent-based model to explore the impact of installers of PV panels on the adoption rate of PV technology among residential households. PV installers and PV household buyers are modeled as agents and the explicit goal is to maximise the profits for panels installers—hence their work is not directly comparable to our methodology. However, a shared aspect is the use of interviews with the involved stakeholders (investors and homeowners) in order to devise the proper (realistic) decision algorithm of the model's agents.

Lee et al. [39] propose to combine ABMs with a logistic regression model to estimate the correct values for the model parameters. They apply this hybrid scheme to the case study of rooftop PV panels adoption in a neighborhood located in Seoul, South Korea. The agents in the ABM are buildings placed in a geographically accurate simulated world, thanks to a geographical information system (GIS). The house owners of the corresponding building decide whether to install a PV panel or not, depending on multiple factors (economical, social, geographic). The parameters of each agent are tuned using logistic regression and very fine-grained building-level data collected during multiple years. The validation is performed using the cumulative observed data (sum of all adopted PV systems in the neighbourhood). In this way, this work uses the observed behaviour to obtain realistic models (the logistic regression is guided by historical data), although the decision algorithm of the agents is not based on self-reported behavior. Moreover, the validation is performed on the same time period used for tuning the parameters (albeit at different granularity), and thus no indication is provided on the predictive capacity of the approach.

While there are recent works that strive to bridge the gap between self-reported and observed behavior (for instance [37,39]), they do not explicitly frame the problem in these terms, thus they only

consider a partial aspect of the overall issue. For example, Lee et al. [39] do not consider self-reported behaviour. Furthermore, the majority of the approaches terminate their analysis at the first step of the validation phase: the ABM parameters are tuned using historical, observed data but no study on the prediction capability of the model is performed. In this way, these approaches are proven to be well suited for describing the observed data (a worthy task), but no guarantee is given about the usefulness for predicting future trends—which is a very important aspect for policy makers. On the contrary, our approach advances the state of the art in two ways: (I) it explicitly states the need of considering both self-reported and observed behaviour, as only via merging them it is possible to obtain accurate ABMs; (II) it is validated on a predictive task, that is the model parameters are tuned using a subset of the observed data while the test is performed using a separate subset (a different time period).

## 3. Methodology Overview

This section provides an overview of the proposed methodology. The rest of the paper is devoted to describe how the proposed approach has been applied to the photovoltaic adoption in the Emilia–Romagna region. Figure 1 depicts the scheme of the methodology. The main idea was to start from the self-reported behaviour (collected through questionnaires and interviews) and extract drivers and barriers influencing the stakeholders' decision process. These decision factors were encoded in an agent-based model via a set of parameters—the relative values of the parameters indicated the magnitude of the impact caused by the associated factor. This ABM was a template for the decision process inferred from the extracted drivers and barriers.

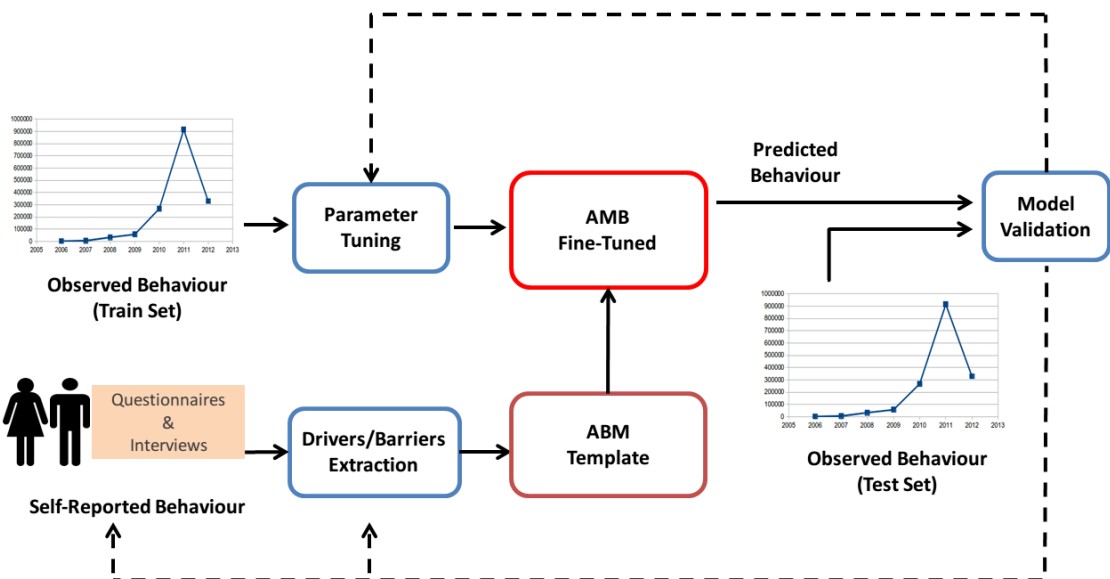

**Figure 1.** Methodology scheme.

A key part of an ABM was assigning a value to the model parameters. Since it was very hard to deduce appropriate parameter values only through self-reported behaviour, the model template underwent a fine-tuning phase where the parameters are empirically tuned using part of the observed behaviour. The observed behaviour was a time series describing the emergent overall behaviour that we want to simulate. Adopting standard machine learning terminology, we partitioned the observed behaviour into training and test sets. We trained the model (changing the parameters' weights) to make the output of the simulator as close as possible to the training observed behaviour and we validated it on the test set.

After the fine-tuning (also referred as calibration in the literature) the final outcome was a simulator that is able to predict with great accuracy the behaviour of interest. Now, the predicted behaviour and the test set (a subset of the observed behaviour) can be used to assess the quality of the

approach; we defined this phase as model validation. The model validation stage can have different outcomes, depending on the accuracy of the predicted behaviour measured using the validation error (the difference between the predicted and the observed behaviour). If the accuracy was not high enough, it was possible to recalibrate the ABM parameters, for example using a different training set or improving the calibration method itself (i.e., letting the fine-tuning algorithms run for a longer time, thus exploring more parameters configurations). If the validation error is too big, a simple recalibration would probably not suffice; in this case the ABM template should be revisited in order to better encapsulate the drivers and barriers previously identified. If the validation results are very poor the best recourse would be a major overhaul of the ABM, possibly repeating the questionnaires and interviews phase with different questions, in order to get a deeper grasp of the agents decisional process. This second refinement should be performed by domain experts and it is out of the scope of this paper.

A key element of the proposed approach was the combination of agent-based models with automated fine-tuning techniques, to calibrate/validate the models. This way led to a method capable of filling the gap between observed and predicted data. We therefore merged methodologies belonging to both agent-based modelling and automated parameter tuning, moving towards a research area that has been very rarely explored so far. In our case, applying an automating tuning mechanism means exploring the configuration space of the ABM with the goal of finding the parameters with the best performance, that is those parameters that minimize the distance between the predicted behaviour and the observed one. As shown in Section 7, an ABM validated using historical data can be also used for predictive tasks with good accuracy, hence helping policy makers devising policies and strategies.

## 4. Driver and Barrier Extraction

The first challenge that needs to be addressed consists of the definition of the agents behaviour. The factors (drivers and barriers) influencing the decision-making process of the actors of the simulation model need to be identified. Drivers are those elements that lead to making a specific energy efficiency investment, while barriers are those elements against the investment. These factor were extracted from a large set of empirical data, gathered through questionnaires and interviews conducted in the Emilia–Romagna region as part of the ePolicy European research project [40]. The empirical data collection took place between March and August 2013. In this section we provide a partial overview of the methodology and the results of the surveys; for the complete and detailed discussion we refer to the work of Balke and Gilbert [41].

The tools used to collect the data are the following:

1. an online questionnaire concentrating on the general attitudes towards photovoltaic and its adoption (196 questionnaires were completed);
2. semi-structured interviews with both apartment block caretakers (building administrators) and employees of photovoltaic installation companies (11 interviews of average length between 60 and 90 min).

The results of both questionnaires and interviews indicated that a series of economic and social elements come into play when deciding to install a PV panel. Some of the factors involved in the decision process were purely economical (for example the profitability of the investment) or physical (such as the type of house and/or the availability of enough roof surface). Other elements reflected personal values and motivations (such as environmental awareness) and represented the social component of the decision process (for example the diffusion of PV-related information within peer networks). A very important aspect to be considered in the simulation is that many households pondering about PV installations are not able to make a clear costs-benefits calculation, but rather act on (other) perceived benefits of photovoltaic.

Some of the drivers identified are the following:

- high energy costs and potential saving after the PV panel installation,

- a general interest on renewable energy and sensitivity towards environmental concerns,
- the possibility to generate their own electricity,
- the knowledge regarding PV technologies (its costs, the incentives, etc.)—often acquired through social networks.

Conversely, these were some of the main barriers:

- misinformation about the cost of PV panels (often perceived as higher than the real one),
- uncertainty about the bureaucratic procedures,
- poor knowledge regarding the incentive mechanisms offered by government bodies,
- fiscal limitations (for example, the up front initial price of the panels),
- trust issues such as the stability of the price for solar-produced electricity in the long run,
- personal low risk propensity.

As illustrated in the following sections, the drivers and barriers identified via interviews and questionnaires form the basis of the decision model of each agent in our simulation. However, even when drivers and barriers have been identified, it is not clear how these parameters are weighted and which is their relative importance in understanding the decision making strategy of different house owners. For this purpose, the model parameters associated with the decision factors will undergo an empirical tuning phase exploiting the observed data, the historical installation rate of PV panels in the Emilia–Romagna region.

## 5. The Agent-Based Model

The process of extracting drivers and barriers from self-reported behaviours and embedding them in an agent-based model is not straightforward. Self-reported behaviour is not always easily quantifiable and, generally speaking, it cannot be used to directly infer a set of rules defining the decision algorithm of the simulated actors. In recent years, the authors of the current paper experimented with several ABMs with the purpose of better capturing the self-reported behaviour. In [42] a purely economic ABM was proposed, in order to understand the impact of national and regional incentives on the adoption of PV panels by residential homeowners. Each agent takes into considerations a series of economic factors that influence its decision to buy and install a PV panel. The model presented in [29] adds a preliminary social component: the behaviour of each agent is influenced by the decisions already taken by its neighbours and by the perception of PV technology possessed by each agent.

The knowledge gained with the previous works led to the methodology presented in this paper. For example, a very important lesson is that discounting non-economical factors generates ABMs fail to properly reflect the observed behaviour. The critical improvement of the currently proposed approach is the refinement of the agents decisional process and, most importantly, the fine-tuning strategy that lead to a model capable of predicting the future trends of PV panels installation rate. A preliminary version of the model discussed here (but without the optimal fine-tuning and predictive capability) was already presented in [43]. The model presented in this paper also adds an entirely new aspect compared to previous works, namely the adoption of real geographical data to obtain a more realistic ABM.

The model discussed in [43] was used mainly as a proof-of-concept of the proposed approach, namely it allowed to explore possible methods to merge self-reported behaviour (encapsulated in the decision algorithm of the agents in the model) and observed behavior (the real, historical data on PV installed power in the Emilia–Romagna region). The agent-based model was created and its parameters were tuned using methodologies described in the following section, but after the fine-tuning no validation was performed. In particular, the model was trained using the historical data gathered in the 2007–2013 time frame; no assessment of its predictive capacity was performed, in part due to a lack of sufficient observed data. Conversely, the current work provides a comprehensive

evaluation of the overall proposed methodology and of the predictive capability of the approach. In the last years, additional observed data was collected (4 more years, 2014–2017) and used to properly validate the ABM, after the parameter tuning.

The model was composed of two types of agents: house owners and the region. The region agent provided regional incentives to house owners; at the start of the period there are some initial funds and each year the region receives a further constant budget to foster the installation of PV plants. The house owners (also referred to as households) are the main actors in the simulation: each house owner decides whether to install a PV panel or not, based on a decisional process illustrated in the following sections. Each house owner is described by a set of attributes: age class, education level, income, family size, consumption, roof area, budget, geographical coordinates and social class. These attributes cooperate to define the household behaviour and to build a social network linking all agents.

The ABM was built in two stages. Firstly, the simulation environment was set up and the virtual world was populated with agents (the households)—this was the configuration phase. The placement of houses followed the actual buildings distribution in the Emilia–Romagna region, in particular taking into account houses positions and their roofs. Then, the social network among household agents was built; the network was created depending on the reciprocal, physical distance between households and the distance in terms of attributes such as class, income, age, and so on. The social network's main contribution was defining how the information about PV system was spread across the simulated world and its agents.

After the configuration, a second stage takes place, called simulation phase in the rest of the paper. During this stage, the simulated world comes to life and the agents begin to ponder whether to install a PV panel or not. The simulation itself can be decomposed as a series of smaller steps, each of them lasting for six months. The installation decision was influenced by several factors, ranging from financial considerations such as the household income and the initial investment cost (and related payback time), to other aspects such as the environmental sensitivity and the neighbours behaviour (neighbours in the social network). The influence of these different factors was encapsulated in four expressions (also referred to as utility functions; these four expressions were then combined in order to establish the desire level of each agent—if the desire remains below a certain threshold, then the household does not install a PV panels, otherwise it proceeds with the investment.

## 5.1. Configuration Phase

The virtual environment initial conditions are defined during the configuration stage; moreover, the simulated world was populated with the agents, placing the buildings and assigning them a roof size according to the actual distribution observed in the Emilia–Romagna region (data made publicly available by the region itself). First, the world-area was filled with buildings and their roofs (fixed geographical coordinates). Secondly, the families-households were created (as many as specified by an input parameter) and each one was placed inside a building (buildings are not shared); households with higher income and the more numerous ones get the buildings with larger roofs.

The positions of the buildings in Emilia–Romagna were obtained by parsing the Ersi shape-files publicly available (http://dati.emilia-romagna.it), which are the results of territorial surveys conducted by the region; in these files each building was represented by a polygon encapsulating multiple information about the building. The agent-based model proposed here requires only position (spatial coordinates) and roof size, hence only these relevant information were extracted. QGIS [44] (an open source Geographic Information Systems, GIS) was the tool employed to parse the Ersi files and collect the needed information (position and roof size).

An important aspect that had a strong effect on the adoption of PV panels (and innovation in general) is the presence of incentive mechanism, aimed at fostering the diffusion of new (or less known) technology. From 2006 to 2014 the Italian government offered national incentives to private citizens willing to install PV panels, namely feed-in tariffs referred to as *Conto Energia*. There have been a few different tariff schemes during the incentives years ([45–48]), differing for the price guaranteed for

the produced electricity. The national incentives are available to all house owners and the tariffs are those actually offered during the considered period. On top of the Italian feed-in tariffs, regional policy makers in Emilia–Romagna devised a number of different additional regional incentive mechanisms, such as investment grants, fiscal incentives, loans, interest funds, etc (see [42] for more details). Historically, the national incentives outweighed the regional ones by at least one order of magnitude therefore their influence has been much stronger [49,50], thus regional incentives have little or no impact on the agents behaviour.

### 5.1.1. Social Classes

The agent households were characterized with a set of attributes, whose values permitted to define the category of the family; these attributes were: income, size (number of components), education level, age class (average value among all components), number of earners, yearly energy consumption and social class. The attribute values assigned to the households followed the real distribution in the Emilia–Romagna region, obtained from the Survey on Household Income and Wealth (SHIW) provided by Bank of Italy (https://www.bancaditalia.it/statistiche/indcamp/bilfait/). On the basis of its attribute values, each agent possesses an associated budget for installing a PV panel. Clearly, the spending capability of a household is directly related to its income class, and this influences the price that each family is willing to pay for purchasing the PV panel. Households with higher income will accept to pay more for the initial investment, while a lower income was associated to a lower budget to invest in a PV system. Generally speaking, households that belonged to the same category (class) made similar decisions when deciding whether to install a PV panel. In practice, the proposed model assumed that this available budget of each family was determined by the attributes defining the family category. A linear regression model was then used to correlate the budget to the explanatory variables obtained from the SHIW data.

Households social classes serve to mimic the different adoption rates of innovative technologies observed in many scenarios [51]. For instance, the so called S-shape curve [52] has been widely used to describe the adoption of an innovation: initially, the adoption rate of a new technology is slow since it is not well understood and its benefits are unclear (or not fully perceived). In a second phase, the adoption rate rises together with the spread of the technology and associated knowledge (mass market phase). Beyond a certain point, the market gets saturated and the adoption rate flattens. Rogers [52] identifies five categories of different adopters: (1) innovators, (2) early adopters, (3) early majority, (4) late majority and (5) laggards. The different categories were usually reflected on characteristics defining each adopter, such as socio-economic status (i.e., high-income individuals can afford to invest more on new and not yet well-established technologies). Each house owner fell in one of the five adopters categories, depending on three of its attributes (the most important features): age class, education level and income. *K*-means was used to identify the five clusters and group the agents belonging to the same class.

### 5.1.2. The Social Network

As discussed previously, the behavior of a household-agent is significantly impacted by its social network. For this reason, during the configuration phase the social links between agents are created, namely each family has a set of friends (other households). Since previous research has shown (see [53–56]) that a small-world topology maps well the real network of relationships that exists between people, the social network adopted in this ABM has small-world properties. Small-world networks are characterised by a shortest-path distance between nodes that increases relatively slowly as a function of the number of nodes in the network [57].

The extended version of the rank-based model proposed by Liben–Nowell et al. [58] was used to get the small-world properties. The probability that a link between node $u$ and node $v$ existed was proportional to a ranking function which depended both on the geographical proximity of the nodes (physical neighbours) and on the attribute proximity of the nodes (how the nodes are similar w.r.t.

their attributes). After a network was built using the extended rank-based method, randomness was added through long-range links. These links drastically reduced the average path length because they connect distant parts of the network. The randomization process takes every edge and rewires it with an empirically obtained probability $p$.

*5.2. Simulation Phase*

In the simulation phase the system evolved as previously described: the decision regarding the installation of new panels took place between 2007 and 2013, then the simulator ran until 2036 to consider the lifetime of PV panels. As described at the beginning of Section 5, each agent had a particular desire level that encapsulates its willingness to invest in a PV panel. The desire level of each household is computed during the configuration phase (Section 5.1), depending on the agent's set of attributes and mathematically expressed as an utility function. The function is a weighted combination of different factors: household income, payback period (of the initial investment), perceived and expected environmental benefits, and the pressure from neighbours (as identified by the social network among agents). The weights were used to combine these factors depending to the household class, or category (see Section 5.1.1). The actual values of the weights cannot be analytically obtained and were instead tuned via model calibration, exploiting the real historical data about PV power installed in the Emilia-Romagna region in the 2007–2013 period (detail in Section 6).

The average lifetime of a PV system was 20 years; the expenses and gains cumulated during this lifespan served to estimate the return on equity (ROE) of a PV panel. The yearly cash flow was computed by subtracting the yearly total expenses from the yearly earnings—clearly considering only PV-related financial movements. The yearly expenses can be obtained by summing the cost of the system divided by its lifetime (mortgage payment), the maintenance costs and the interests on eventual loans. Potential yearly earnings comprised the surfeit electricity sold to the national electrical grid and the electricity bill savings granted by self-production. Alongside, there could be national and/or regional incentives, with a profound impact on the overall profitability of an investment. The incentives can influence yearly cash-flows in different ways: for instance, the gains are directly linked to the Italian national feed-in tariffs, while yearly expenses depend on the initial cost (affected by regional investment grants) and loan interests (target for several incentive schemes).

Each household had to find the optimal size for the PV system (the size that maximises the ROE); if the set of conditions characterizing an agent were unfavourable, the house owners can also opt not to install a PV panel. The problem was solved with an heuristic algorithm based on Simulated Annealing [59]. The proposed model assumed that households aimed at making well informed decisions, for example by getting advice from PV installers in order to properly understand the available options. Hence, agents are supposed to purchase PV panels with the goal of maximising their reward w.r.t. energy production and financial savings.

The Utility Function

One of the most important component of the proposed approach is the criterion used by agents/households to decide whether to install a PV panel. As mentioned earlier, this decision is taken by each agent based on the values of its attributes. The decision criterion for agent $v$ is expressed by the utility function (also referred to as desire level):

$$U(v) = w_P(c_v)u_P(v) + w_B(c_v)u_B(v) + w_E(c_v)u_E(v) + w_N(c_v)u_N(v) \qquad (1)$$

where $c_v$ is the class of the agent (details in Section 5.1.1). The utility function is a weighted combination of four components:

1. the investment payback time $u_P(v)$, representing the expected payback period of the PV panel;
2. the available budget $u_B(v)$, that strongly impacts the possibility to make the initial investment (without considering external incentives);

3. the impact of the neighbours' choices $u_N(v)$;
4. the potential benefits generated by investing in a PV panel $u_E(v)$ (estimated in relation to a decrease in consumption of electrical energy from other non-renewable sources).

Each factor in Equation (1) was weighted by a class-dependent parameter; these weights are $w_P(c_v)$, $w_B(c_v)$, $w_N(c_v)$ and $w_E(c_v)$—the notation serves to express their dependency on the agent class $c_v$. Each agent had its set of specific weights; agents of the same social class do not share the same weights (at least not necessarily—this still could happen as a byproduct of the parameters tuning procedure). A key aspect of the proposed approach was assigning correct values to these weights: this is the crucial operation where the self-reported behaviour obtained with questionnaires and interviews (which guided the definition of the agents behaviour) was merged with the observed behaviour (historical data of installed PV power). This passage will be described in detail in Section 6.

The first factor in Equation (1) regards the payback period, $pp$. To obtain a balanced influence of all factors in the utility functions, all factors where normalized in a [0,1] range. In the case of the payback time, the normalization takes advantage of the bounds on the minimum payback period $min(pp)$ (assumed to be equal to one year) and on the maximum payback period $max(pp)$, assumed to be 21 years since the expected useful life for PV systems is 20 years. Hence, the payback influence for agent $v$ is computed following [28] and expressed by this equation:

$$u_P(v) = \frac{max(pp) - pp(v)}{max(pp) - min(pp)} = \frac{21 - pp(v)}{20} \tag{2}$$

where $pp(v)$ is the payback period for the initial investment. Its value is computed using the net present value (NPV) of the PV system—the NPV typically starts with negative values (due to the initial cost of the investment) and it gradually gets closer to zero, while the initial cost is offset by yearly gains due to electrical bill savings and sale of own-produced energy. When the NPV turned from negative to positive it indicated the point when the investment became profitable. The computation of the NPV was based on the yearly cash flows—each agent measures its expenses and gains (taking into account also national and regional incentives) and computes its yearly NPV accordingly.

The household budget $u_B(v)$ is given by:

$$u_B(v) = \frac{e^{v_{budget}}}{v_{equity}} \tag{3}$$

with the initial investment $v_{equity}$ computed as the PV panel installation cost minus any applicable incentive. $v_B$ is the disposable budget of the household.

The third factor contributing to the agents' decision is the environmental benefit that can be gained by adopting PV technology, instead of consuming electrical energy coming from non-renewable resources. These benefits are measured in terms of oil saved, which is in turn correlated with an overall decrease in $CO_2$ production. The Italian Regulatory Authority for Electricity and Gas provides a factor to convert the produced energy (expressed in MWh) to the equivalent in tonnes of oil (TOE) (A TOE is defined as the amount of energy released by burning one tonne of oil, or 0.187 TOE for each MWh produced). Thanks to this conversion, the ecological benefits can be computed with the following equation:

$$u_E(v) = \frac{1}{e^{oil_{notConsumed} - oil_{consumed}}} \tag{4}$$

The final component of the desire function (Equation (1)) is the influence of the other members of the social network of the agent. This factor is identified by $u_N(v)$ and it encapsulates the importance of the neighbours' choices in shaping the household behaviour. As previously mentioned, the agent's neighbours are the nodes (other households) with a shared links; the vicinity of two nodes depends

on geographical proximity and social class similarity. The neighbourhood influence contribution is computed with the following equation:

$$u_N(v) = \frac{1}{1 + e^{\frac{1}{2}L_{v,tot}L_{v,adopter}}} \tag{5}$$

with $L_{v,tot}$ being the total number of links of agent $v$ and $L_{v,adopter}$ the number of links shared with adopters.

## 6. Parameter Tuning

So far, only the agent-based model has been described. The model has been built upon the insights gained by analysing the self-reported behaviour, gathered through questionnaires and interviews. By leveraging this information it was possible to identify barriers and drivers that affect the decision criterion regulating the installation of a PV panel (see Section 4). These factors were then used as a guide for the behaviour of the agents in the simulation world; however, the algorithm describing the agents' behaviour hinged on a set of parameters that cannot be easily obtained through analytical tools. At this point, the second stage of the proposed methodology came into play, namely the observed behaviour. Historical data will be used to fine-tune the model parameters, thus obtaining a model which is capable to faithfully describe real world dynamics and that can be used to make accurate predictions, as reported in Section 7.

The historical data of the PV panels installation trend in the Emilia–Romagna region was gathered looking at the data provided by the Italian government [49], in particular, the PV installation trend in Emilia-Romagna from 2007 to 2017 (Unluckily the data regarding years earlier than 2007 is very scarce due to the almost negligible consideration given by the Italian government to PV technology). The data set is divided in two chunks: training set, from 2007 to 2013, and test set, from 2014 to 2017. The training set is used to fine-tune the model parameters (trying to fit the simulated trend to the observed one). Afterwards, the trained model can be used to predict the PV installation rate during the test period; then, it is possible evaluate the quality of the prediction and thus the accuracy of the model, by comparing the historical data with the predicted one.

As a reminder, the parameters that needed to be tuned were the weights of the utility function: $w_{pp}(c_v)$, $w_B(c_v)$, $w_N(c_v)$, and $w_E(c_v)$. In practice, the scope was to find the weights values that better fit the curve representing the Emilia–Romagna PV power installation rate. The tuning problem can be seen as fitting a model to real data; there exist several methods to perform this task. After a preliminary evaluation of different methods, a genetic algorithm (GA) [60] came across as the technique that provided best results without requiring excessive computational resources. This happens because GAs are apt at finding solutions in spaces where it is hard to derive analytical models and it is not easy to mathematically find global optima. Another benefit derived from the use of a genetic algorithm is the fact that they have been proven to be very effective at dealing with problems where small changes in the weight configuration can lead to a great impact on the final outcome. This is the case of the proposed agent-based model: the four factors of the desire function are strongly intertwined and linearly combined (see Equation (1)). Moreover, the decision criterion is influenced also by the social interaction: the weights assigned to a given agent can modify its decision, which in turn has an impact on the decision process of other (possibly many) neighbours.

The genetic algorithm began with a random initial population of parameters configurations (the "individuals" in GA terminology). The initial population was then evaluated by running the agent-based model and observing the PV installed power by all households, given the currently applied parameters. Since during the training phase, the target PV power is available (the real, historical PV installation data in Emilia–Romagna), it was possible to assess the accuracy of the fit of the current population, by measuring the difference between target and simulated power. After the evaluation, the GA selected the next generation of individuals (a different set of model parameters), which were evaluated as well, with the goal of finding the best fitting population. To generate the next population

the tournament selection [61] was employed: $k$ individuals are selected from the actual population using $n$ tournaments of $j$ individuals. From every tournament emerges a winner (the individual with the highest fitness—the one generating the smallest distance between simulated and historical PV power) and this is the parameter configuration selected for the next generation.

The new population was not deterministically decided with the tournament mode, but a certain degree of randomness is introduced through crossover and offspring mutation. The former mechanism randomly chose two individuals for reproduction and one or multiple children were bred from them; in the proposed genetic algorithm one-point crossover has been used. A single crossover point on both parents' configuration is selected and a new child configuration is created via a swap of the values beyond the crossover point. The second random mechanism, mutation, works by randomly modifying values (parameters of the agent-based model) in randomly chosen individuals. The evolution process (creation of offspring, random mutations, evaluation) was repeated four hundred times.

## 7. Model Validation

After the parameter tuning via genetic algorithm described in the previous section, the resulting model accuracy needed to be measured. For this analysis, the ABM was composed by by 2000 agents; each simulation required around 10 s with a 2.40GHz Intel QuadCore (i7-5500U CPU) with a 16GB of RAM. The genetic algorithm used a population of 50 individuals and the overall time required to calibrate the model was around 30 h. As mentioned before the parameters tuning was made using observed data in the period 2007–2013; observed data in the 2014–2017 range was used only during testing. To summarize, the experimental setup was the following: (1) create an ABM and calibrate its parameters with the genetic algorithm; (2) simulate the 2007–2017 period with the fine-tuned ABM; (3) observe the simulated PV cumulative installed power and adoption rate—the difference between observed and simulated data in 2007–2013 measures the quality of the fine-tuning technique, while the difference measured in the 2014–2017 period serves to evaluate the predictive capability of the model. This scheme was repeated 30 times to obtain statistically significant values; in the rest of the paper, only mean values were reported (in both graphs and table). As metrics for the evaluation, we considered the mean absolute error (MAE), the root mean squared error (RMSE), the mean absolute percentage error (MAPE), and the coefficient of determination ($R^2$). These are standard metrics and are defined by the following equations (the equation for $R^2$ is not reported for the sake of clarity):

$$MAE = \frac{1}{N} \sum_{i=1}^{N} |o_i - s_i| \tag{6}$$

$$RMSE = \sqrt{\frac{1}{N} \sum_{i=1}^{N} (o_i - s_i)^2} \tag{7}$$

$$MAPE = \frac{100}{N} \sum_{i=1}^{N} \frac{|o_i - s_i|}{o_i}, \tag{8}$$

where $N$ is the number of runs (30), $s_i$ is the simulated value, for instance the PV power installed in one year in the simulated environment of the ABM, and $o_i$ is the observed value.

We considered two outcomes to measure the fine-tuning and prediction results: (1) the yearly installed PV power and (2) the cumulative installed power, that is the total value obtained by summing previous years installed power and the current year's installed capacity. The former value better reflects the yearly changes in adoption rate while the latter could be more useful to policy makers to devise strategies for reaching long-term goals (i.e., a certain amount of total PV power by year 2020). Table 1 reports the validation results after the parameter tuning. Each row corresponds to a year; the table includes both the period used as training set (2007–2013) and the period used to evaluate the predictive capability (2014–2017), separated by a horizontal line. The last row reports the average values computed over the whole time frame 2007–2017. The first column corresponds to the year;

the following three columns report MAE, RMSE and MAPE for the yearly installed PV power; the last three columns show the same metrics computed on the cumulative installed power.

**Table 1.** Validation and prediction results of installed photovoltaic (PV) power; yearly rate and cumulative installations.

| Year | Yearly Installed Power | | | Cumulative Power | | |
|---|---|---|---|---|---|---|
| | MAE | RMSE | MAPE | MAE | RMSE | MAPE |
| 2007 | 0.02 | 0.04 | 0.001 | 0.0003 | 0.0004 | 3.33 |
| 2008 | 0.53 | 0.62 | 15.41 | 0.006 | 0.008 | 12.93 |
| 2009 | 0.24 | 0.35 | 4.76 | 0.004 | 0.004 | 3.83 |
| 2010 | 0.37 | 0.35 | 3.03 | 0.006 | 0.007 | 0.26 |
| 2011 | 1.07 | 1.43 | 4.04 | 0.012 | 0.023 | 2.65 |
| 2012 | 0.3 | 0.35 | 1.39 | 0.011 | 0.018 | 1.58 |
| 2013 | 0.3 | 0.41 | 3.04 | 0.015 | 0.016 | 1.83 |
| 2014 | 0.27 | 0.21 | 4.05 | 0.013 | 0.02 | 1.43 |
| 2015 | 1.74 | 1.82 | 32.95 | 0.012 | 0.018 | 0.51 |
| 2016 | 0.44 | 0.64 | 7.55 | 0.0002 | 0.0012 | 0.07 |
| 2017 | 0.51 | 0.72 | 8.59 | 0.017 | 0.021 | 1.68 |
| *Average* | 0.52 | 0.69 | 7.71 | 0.008 | 0.009 | 2.73 |

Let us consider first the yearly adoption rate. Following the type of validation made by most other works in the literature, that was looking only at the results on the training set, it can be noted that the fit was extremely good. The average $R^2$ on the training set was equal to 0.997, the MAE was equal to 0.385, the RMSE was equal to 0.483, and the MAPE was equal to 4.52. These values indicate that the parameter tuning was very effective. A higher MAPE can be observed for the year 2008 (15.41); the simulated installed PV power was lower than the observed one and this could indicate that the current version of the ABM did not include some drivers that boosted the adoption of PV panels in early years. Moving on and considering also the predictive capability of the proposed approach, the results remain promising. However, it can be observed that the accuracy decreases in the test set, especially in 2015, where the simulated installed PV is significantly lower than the observed one; the proposed ABM overestimates w.r.t. the historical data. The statistical results obtained considering both the train and the test set were the following: $R^2$ = 0.991, MAE = 0.524, RMSE = 0.693, and MAPE = 7.71. Conversely, if we computed the average values only over the test period, these were the results: MAE = 0.896, RMSE = 1.16, and MAPE = 16.36. The precision was skewed especially by the large error made in 2015 (MAPE equal to 32.95). Probably this was due to the fact that the ABM underestimates the impact of the reduced national and regional incentives on the households decision process; as a reminder, national incentives by the Italian government ceased at the end of 2013.

In order to contextualize these values, the comparison with the validation results proposed by a recent related work could be considered, namely Lee et al. [39] (see Section 2 for details); this work was chosen because it provided easily comparable metrics and has a similar approach (tuning the parameters of a ABM using historical data). The comparison might not be completely fair since Lee et al. used different sets for parameter tuning and validation, respectively, building-based data and area-wide (summing the power of all PV panels installed in a neighbourhood) data while our ABM was trained and validated using region-wide yearly adoption data. However, the comparison is legitimate because we include the test set not used for the parameters calibration, thus increasing the difficulty of the task tackled by the proposed approach. Lee et al. proposed three different ABMs, with MAPE equal to 14.86, 13.57, and 62.52 (average value computed over all years). Our ABM instead has an average MAPE (considering both training set and test set) equal to 7.71, a significantly lower value. Moreover, it can be worth to notice that even the results on the test set alone (MAPE = 16.36) are similar to those obtained by Lee et al., although these numbers are not really comparable since they

refer to entirely different tasks, pure validation (Lee et al.) and prediction (the approach proposed in this paper).

Figure 2 shows the PV installation growth rate in the considered time period (2007–2017). The solid blue line corresponds to the PV growth rate predicted by the agent-based model while the black dashed lines depicts the real installation trend in Emilia–Romagna. The *x*-axis displays the year and the *y*-axis reports the PV power growth in percentage. The figure contains the results for both the training set (period 2007–2013) and the test set (2014–2017). In this way it is possible to observe the quality of the proposed fitting mechanism, by observing the lines discrepancies in the training set. At the same time, the figure reveals that the ABM does not suffer from overfitting and the proposed methodology can be used to create models that generalize well and, consequently, that can be used for predictive purposes—by looking at the differences between the lines in the test set. The visual analysis revealed results similar to those presented with the quantitative analysis. In fact, it can be noted that the parameter tuning was very effective (in practice the two lines overlap in the training set) and that the predictive capability are good as well, albeit slightly less accurate. Focusing on the PV case study, there were also useful insights that can be gained by looking at the installation trends, both real and simulated. Both curves clearly indicated that the initial growth rate has been relatively slow (2007–2009), possibly motivated by an initial reluctance due to limited knowledge and doubts about the PV technology. As the knowledge gets more widespread and the Italian national incentives increased (higher feed-in tariffs were offered in the years 2010 and 2011), the installation growth increases steeply, with a distinct peak around 2011. After 2011, there was a steady decline in number of new PV panels installed, as the financial benefits for homeowners start to become smaller, due to a decrease in national incentives not compensated by regional incentives or by a sufficient decrease in the cost of the technology.

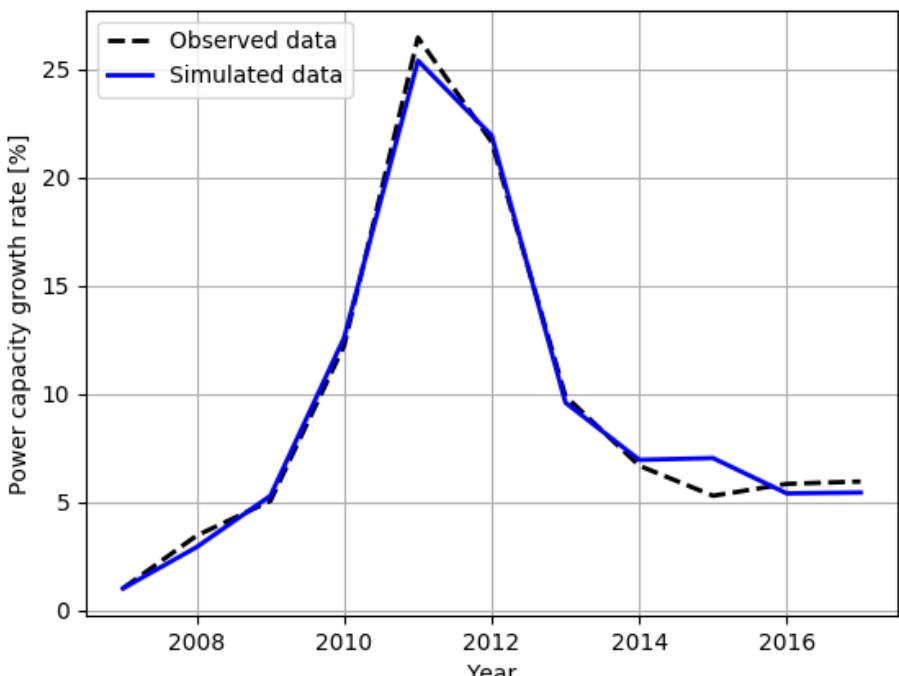

**Figure 2.** Model calibration results—yearly installed photovoltaic (PV) power.

This could be partially explained by the extremely complex situation that occurred in the Italian PV technology field in the last few years, with longstanding incentive mechanisms that abruptly came to an end and different regulations following one another. This situation generated a marked discontinuity in the installation trend, a discontinuity that was very hard to forecast. Clearly, there is still room for improvement in terms of ABM accuracy, but it is important to notice that the proposed

approach can already emulate the observed behaviour with a precision more than sufficient to help policy makers in their decisions.

Now we look at the validation results with the cumulative installed power (last three columns of Table 1). The effect of the worse accuracy computed in the test set is amplified by the smaller magnitude of the installation rates observed in the 2014–2017 period w.r.t. the peak values observed in previous years. If we consider the total PV installed over the years the results of the parameters calibration are still very good and the "small-values" effect noticed in the test set loses its influence. Figure 3 shows the cumulative installed PV power in Emilia-Romagna in 2007–2017, both according to the historical data (black dashed line) and to the agent-based model (red continuous line). Since our simulator considers only 2000 agents against the millions of households in Emilia–Romagna, the historical and simulated absolute values differ by orders of magnitude; in order to render the comparison possible both sets (observed and simulated) were normalized dividing by the maximum value (year 2017).

As both the quantitative analysis and the graph reveal, the parameters tuning works even better for the cumulative installed power. In this case, the average values computed over the whole time frame were the following: $R^2 = 0.999$, MAE = 0.008, RMSE = 0.009, and MAPE = 2.73. The error was lower w.r.t. the case of the yearly PV installed power because the yearly changes in adoption rate are "smoothed" by the sum operation. This effect is even more pronounced when considering the test set alone. In this case the MAE is equal to 0.011, RMSE equal to 0.015 and MAPE equal to 0.92; the MAPE in particular was even lower than the training set case—this happened because in the last years (2014–2017) the PV panels installation rate has greatly slowed down, hence the errors made with the years in the test set have a relatively smaller impact on the total installed power.

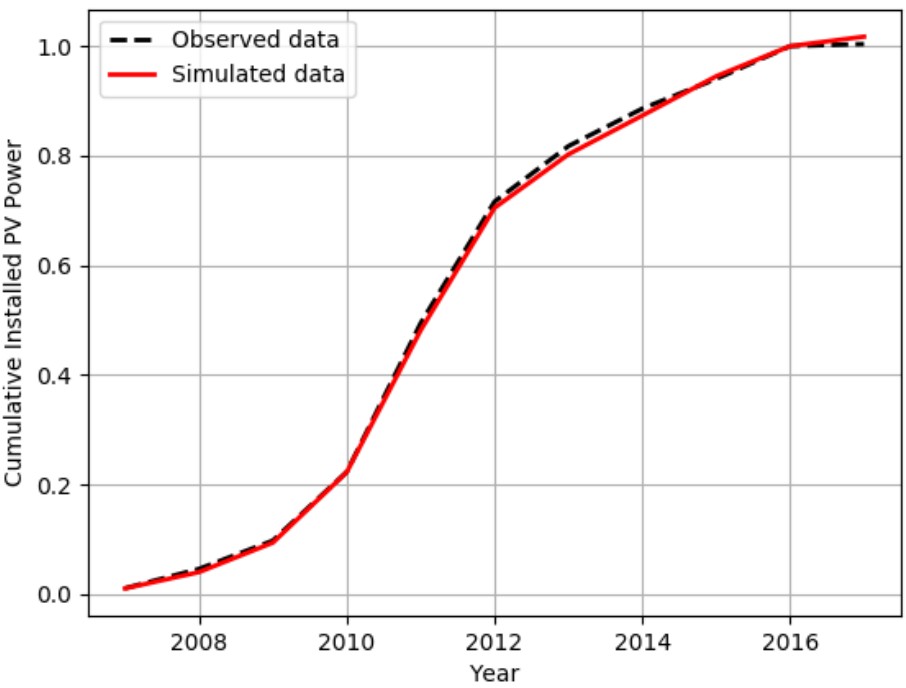

**Figure 3.** Model calibration results—normalized cumulative installed PV power.

## 8. Conclusions

In this paper we presented a novel methodology to fill the gap between self-reported behaviour and observed behaviour, by means of agent-based model and empirical parameter tuning. As a case study, we considered the diffusion of photovoltaic power in the Emilia–Romagna region of Italy. The first step of the approach consisted in a data collection phase to enable the identification of the drivers and the barriers that influence the decision making process of house owners faced with the

possibility of installing a PV panel on the top of their houses. The data have been collected through online questionnaires and interviews.

These drivers and barriers were then used to model the decision process of the agents composing the simulation. Having both self-reported and observed behaviour, parameters attached to the various decision factors can be empirically tuned, thus enabling agent-based models to be used also for predictions, even in an approximate manner. The idea is that given the agent-based model based on the self-reported behaviour, its parameters can be adjusted and tuned exploiting past real data. Hence, a ABM that takes into account economic, social and geographical factors to emulate the self-reported behaviour has been proposed. The model is characterized by a set of parameters that were fine-tuned using a Genetic Algorithm. Finally, the accuracy of the model prediction has been evaluated, by analysing the difference between the historical PV installation rate and the results produced by the simulator. The results are very promising and the proposed approach can be used by policy makers to guide their decisions.

The future research directions that have yet to be explored are the following. First, the ABM can be refined in order to achieve a even greater prediction accuracy. Second, it is important to test the proposed methodology in different conditions (different region/countries, extended time period). Another possible direction to explore consists of scaling up the simulation size, up to the point of including hundreds of thousands of agents; this should lead to result even closer to the observed data. In our opinion, the most promising direction is to integrate the proposed approach and predictive model in a larger scheme aimed at helping policy makers with their task. After having bridged the gap between self-reported behaviour and observed behaviour, the following step would be to reach a target behaviour, i.e., the desired level of photovoltaic power production. For this purpose, the agent-based model could be used to extract the best guidelines for policy makers to achieve the desired strategic objectives.

**Author Contributions:** Conceptualization, A.B. and M.M.; data curation, A.B.; funding acquisition, M.M.; investigation, A.B.; Methodology, A.B.; Project administration, M.M.; resources, M.M.; software, A.B.; supervision, M.M.; validation, A.B.; writing—original draft, A.B.; writing—review & editing, A.B. and M.M.

**Funding:** EU ePolicy project (FP7/2007–2013), g.a. 288147.

**Conflicts of Interest:** The authors declare no conflict of interest.

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
