# Peer review of "Merging Observed and Self-Reported Behaviour in Agent-Based Simulation: A Case Study on Photovoltaic Adoption"

_applsci, doi:10.3390/app9102098_

Round 1
Reviewer 1 Report
This paper introduces a case study of photovoltaic adoption in the Emilia-Romagna region of Italy. The authors present an agent-based model that is tuned according to historical data. For this purpose, the authors propose a GA based approach to adjust the stimulation parameter. The model is evaluated on a test set.
While the paper topic is of interest, it suffers from further limitations:
- The contribution and its novelty are not clear. While the state of the art focuses on modelling photovoltaic adaption, the authors state that the contribution is the fine-tuning of their agent-based model. Model calibration on agent-based systems is a rich topic in scientific literature and therefore, the state of the art does not help to position this contribution in regards to recent work in this domain. In its current form, the state-of-the-art is more a listing of past approach and lack of critical analysis to highlight the problem addressed. Furthermore, most of the referenced work is prior to 2015, whereas recent work has been done on the photovoltaic adaption domain (Lee, M., & Hong, T. (2019). Hybrid agent-based modelling of rooftop solar photovoltaic adoption by integrating the geographic information system and data mining technique. Energy Conversion and Management, 183, 266-279.| Adepetu, A., Alyousef, A., Keshav, S., & de Meer, H. (2018). Comparing solar photovoltaic and battery adoption in Ontario and Germany: an agent-based approach. Energy Informatics, 1(1), 6.).
- More than half of the paper is a word-to-word cut and paste from previous authors work (Iachini, V., Borghesi, A., & Milano, M. (2015, September). Agent-based simulation of incentive mechanisms on photovoltaic adoption. In Congress of the Italian Association for Artificial Intelligence (pp. 136-148). Springer, Cham.). It looks like there is no novelty in the presentation of the model, whereas a better formalisation and schematics might have helped to understand it. The novelty of the contribution is not clear and no comparison with results from the previous implementation is given. It gives the impression that this paper is an extended version from the previous one, but it is not clear what novelty it brings.
For those reasons, it is difficult to accept this paper in its present form, as a better positioning regarding the state-of-art is needed, and a clear highlighting of the contribution, including a comparison with author’s past approach, is missing.
Author Response
We thank the reviewer for the useful comments.
- We revised the "Related Work" section in order to clarify our work's position
and to better highlight our contributions.
- We revised entire sections describing the proposed approach and the results
obtained -- old parts are still available in the revised manuscript but they
have been struck out (crossed with horizontal lines); the new text appears in
bold (to improve clarity, we also provide a revised version without the old
parts but containing only the revised content).
- The main differences with the work of Iachini 2015 are the following:
1) The overall proposed methodology, encompassing the merging of self-reported
behaviour with observed behaviour. This methodology is particularly important
as, in general, computational social science bases behavioural models only on
surveys and interviews collecting self-reported behavior, while we show that by
having data on the real observed behavior we can learn and extract a more
accurate model of the phenomenon at hand.
2) The section on the drivers & barriers extracted via questionnaires that guide
the creation of the ABM. Drivers and barriers are extracted by self-reported
behavior.
3) A critical validation of the overall scheme: the work of 2015 was used mainly
as a proof-of-concept of the proposed approach, the agent-based model was
created and its parameters were tuned, but after the fine-tuning no validation
was performed. In particular, the model was trained using historical data
gathered in the 2007-2013 time frame. Due to lack of available data at the
time to be used for testing purpose, we could not prove that the proposed scheme
would have fare accurately in a realistic setting. We could only claim that our
methodology was technically sound and it could lead to correctly fitted models.
In this work, we instead perform a proper test of the proposed methodology by
testing our approach and its predictive capability, because I) we extended the
data collection time span adding three more years and II) we validate the
model after the parameter tuning, thus assessing for the first time the actual
predictive capability of the proposed methodology.
Reviewer 2 Report
In this paper Authors presented a novel methodology to fill the gap between self-reported behaviour and observed behavioural, by means of agent-based model and empirical parameter tuning. As a case study Authors considered the diffusion of photovoltaic power in the Emilia-Romagna region of Italy.
The originality the concepts, the significance and the methods are good. The completeness and the organization of manuscript of the paper are good. The organization of the manuscript is good. In my opinion the technical treatment is plausible and free of technical errors.. Below I presented some remarks that came to my mind during reading.
Remarks:
1. According to Applied Sciences template, abstract should not exceed 200 words. Therefore, abstract should be shortened accordingly.
2. Lines 26-27: “Founding” and “Conflicts of Interest” should be place and the end of the paper. Check in Applied Science template and correct this.
3. Lines 85-90: In the structure of the paper, describe all chapters, not only chapters 2, 4, 5 and 8.
4. Paper should be written in an impersonal form.
5. Names of chapters and sub-chapters – each word is written with a capital letter (check in the Applied Sciences template how the names of chapters and subchapters should be described). Please check this throughout the paper and correct this.
6. Equations must be numbered. Check this throughout the paper and correct this.
7. Figure 2 should be placed in Chapter 7.
8. Figure 3 should be placed in Chapter 7.
Author Response
We thank the reviewer for the comments.
We shortened the abstract
"Funding" and "Conflict of Interest" placed in correct position
Changed paper structure description
Done
Sections/Subsections titles changed
Equations are now numbered
Done
Done
Round 2
Reviewer 1 Report
The massive modifications made by the authors have increased the quality of the paper. Therefore, I recommend it's publication in its present form.